# CODE TRANSLATION WITH COMPILER REPRESENTATIONS

**Marc Szafraniec**[*]  **Baptiste Rozière**[*]   **Hugh Leather**   **François Charton**
**Patrick Labatut**   **Gabriel Synnaeve**
Meta AI
{mszafraniec,broz}@meta.com

## ABSTRACT

In this paper, we leverage low-level compiler intermediate representations (IR) to improve code translation. Traditional transpilers rely on syntactic information and handcrafted rules, which limits their applicability and produces unnatural-looking code. Applying neural machine translation (NMT) approaches to code has successfully broadened the set of programs on which one can get a natural-looking translation. However, they treat the code as sequences of text tokens, and still do not differentiate well enough between similar pieces of code which have different semantics in different languages. The consequence is low quality translation, reducing the practicality of NMT, and stressing the need for an approach significantly increasing its accuracy. Here we propose to augment code translation with IRs, specifically LLVM IR, with results on the C++, Java, Rust, and Go languages. Our method improves upon the state of the art for unsupervised code translation, increasing the number of correct translations by 11% on average, and up to 79% for the Java → Rust pair with greedy decoding. With beam search, it increases the number of correct translations by 5.5% in average. We extend previous test sets for code translation, by adding hundreds of Go and Rust functions. Additionally, we train models with high performance on the problem of IR decompilation, generating programming source code from IR, and study using IRs as intermediary pivot for translation.

## 1 INTRODUCTION

Automatic code translation allows to port old codebases to new frameworks, or high-level (but slow) languages to low-level (and fast) ones. Current industry solutions, known as *transpilers* or *transcompilers*[1], rely on handcrafted rules that are applied systematically. They produce unidiomatic translations that prove hard to read for human programmers. This is a serious limitation: the translated code should be easy to read and understand, as it will eventually be maintained by human developers.

In recent years, Neural Machine Translation (NMT) was proposed as an alternative to rule-based code translation (Roziere et al., 2020; Weisz et al., 2021; 2022). These models, trained from existing human-readable code, produce idiomatic, easy to understand, translations. Unfortunately, neural transpilers are unreliable, and often fail to translate the semantics of the input program accurately. This is a serious limitation, as some of the human work saved by the transpiler has to be reinvested debugging its output.

We propose to improve the reliability of NMT by leveraging information from compiler toolchains. When processing source code, compilers create Intermediary Representations (IR): language-agnostic pseudocode that describes the semantics of the program. Augmenting training data with the corresponding IR can benefit a Neural Transpiler in two ways: it helps align embeddings for different languages and improves the semantic understanding of the code. As shown in Figure 1, this can greatly improve the semantic quality of neural translations.

In this work, we leverage LLVM (Lattner and Adve, 2004) to augment source code with corresponding Intermediate Representation and train models for code translation and decompilation. We compare

---

[*]Equal contribution
[1]https://en.wikipedia.org/wiki/Source-to-source_compiler

| Input function | TransCoder | TransCoder-IR |
|---|---|---|

```cpp
// C++
int nthTerm ( int n ) {
   return ( n*n ) + ( n*n*n ) ;
}
```

```rust
// Rust
fn nth_term (n: usize )-> usize {
   ( n*n ) + ( n*n*n )
}
```

```rust
// Rust
fn nth_term ( n : i32 ) -> i32 {
   ( n*n ) + ( n*n*n )
}
```

```java
// Java
static int divisorSum(int n){
   int sum = 0 ;
   for(int i=1; i<=n; ++i )
      sum += ( n / i ) * i;
   return sum ;
}
```

```go
// Go
func divisorSum(n int) (int, int) {
  sum := 0
  for i := 1 ; i <= n ; i ++ {
    sum += ( n / i ) * i
  }
  return sum , n
  }
```

```go
// Go
func divisorSum ( n int ) int {
   sum := 0
   for i := 1 ; i <= n ; i ++ {
      sum += ( n / i ) * i
   }
   return sum
}
```

```java
// Java
static boolean isDivBy9(int n){
   if ( n == 0  n == 9 )
      return true;
   if ( n < 9 )
      return false;
   return isDivBy9(
         (int)(n >> 3) - (int)(n & 7)
         );
}
```

```go
// Go
func IsDivBy9 ( n int ) bool {
   if n == 0  n == 9 {
      return true
   }
   if n < 9 {
      return false
   }
   return IsDivBy9 (int(n) > 3 )
            - int(n & 7)
}
```

```go
// Go
func IsDivBy9 ( n int ) bool {
   if n == 0  n == 9 {
      return true
   }
   if n < 9 {
     return false
   }
   return IsDivBy9 (
         int(n >> 3) - int(n & 7)}
         )
}
```

Figure 1: **Improvements over TransCoder.** The first example shows a translation from C++ to rust, where TransCoder generates code using unsigned instead of signed integers. In the second example, a translation from Java to Go, it generates a function with the wrong return type. In the third example, which is also a translation from Java to Go, the model outputs a function that looks similar to the correct solution but it confuses > with » and closes an expression with a parenthesis too early. In these cases and many others, TransCoder makes mistakes that are small in terms of edit distance, but have a large impact on the semantics of the code. Using the IR to ground the representations to the semantics often helps solving these issues.

it to TransCoder, which uses only code and no IR. We also design an IR-only baseline, dubbed the pivot method, which generates a translation solely by decompiling an IR generated from the source language to a different target language. We experiment with four languages: C++ Java, Rust and Go, and show that utilizing both the code and the IR allows for an average relative improvement of 5.5%. Moreover, our method only uses the IR at training time and does not require extra computations at inference time.

Our main contributions are:

- We implement a new IR-augmented translation method, which leverages LLVM IRs to improve code representations. It allows us to increase the number of correct translations generated by TransCoder for C++, Java, Go and Rust by 5.5%. Compared to our IR-only pivot method, the improvement reaches 170%

- Our method is especially useful in the low data regime: with relative improvements reaching 29.7% when translating to Rust and 25.6% when translating from it.

- We extend the parallel evaluation dataset of 852 functions in C++, Java and Python from Roziere et al. (2020) with 343 more functions in Go and 280 more in Rust, along with corresponding test cases

- In addition, we achieve 78% accuracy when decompiling LLVM IRs to C++

## 2 INTERMEDIATE REPRESENTATIONS IN COMPILERS

Compilers translate programs written in a computer language into executable code for a specific machine. Most compilers consist of a front-end taking source code as input, and a back-end which produces machine binary code. The front-end lexes (tokenizes) and parses the program. Then, it produces an abstract syntax tree (AST), and translates it into some Intermediate Representation (IR). The back-end converts the IR into machine-specific executable code.

In modern compilers such as LLVM (Lattner and Adve, 2004), the IR is generic across different input languages (and thus different front-ends). It allows the application of transformations and target

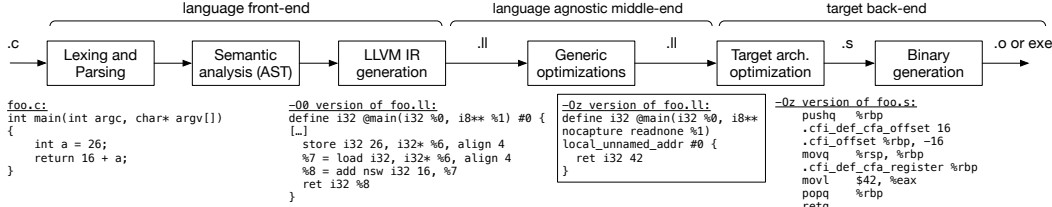

Figure 2: A bird's eye view of a compiler toolchain, exemplified with LLVM. The unoptimized version (-O0) is shown here for illustration. In practice we used the size-optimized version (-Oz) of the IR as boxed, which does the compile time optimization of computing the addition of 26 and 16.

agnostic optimizations to the IR, in a middle-end module independent from the source language and target machine. This results in an efficient compiler structure: new languages can be implemented by rewriting the front-end, and new target machines by rewriting the back-end.

Several IRs usually co-exist in a compiler: each stage in the toolchain (Figure 2) introduces a new representation. Early stage IRs are language-dependent (e.g. ASTs mirror the syntax of the source language). Late stage IRs replace named variables by registers and reflect the specifics of the target architecture. In this work, we are interested in middle-end IRs, which are independent from the target machine, and similar for all source languages (like dialects in natural languages).

## 3 TRAINING OBJECTIVES

Unsupervised machine translation consists of learning multilingual sequence embeddings, and generating sequence in any output language from these embeddings (Lample et al., 2018a). We now present the objective functions for these tasks. In section 3.1, we review the three basic objectives used by TransCoder, our baseline NMT system. In section 3.2, we introduce three new functions that leverage LLVM IRs to improve the multilingual representation of source code, and the performance of our translation models. During training, we alternate between all six objectives, running each for the same number of optimisation steps. At inference, the model is only provided with the source code, i.e. the IR is not needed.

Formally, let $x = x_1 \ldots x_{N_{so}}$ be the source sentence, $z^{(x)} = z_1^{(x)} \ldots z_{N_{ir}}^{(x)}$ the corresponding IR, and $y = y_1 \ldots y_{N_{ta}}$ the target sentence. We write $\mathcal{L}_{CE}(\hat{y}, y) = \sum_i \ell_{CE}(\hat{y}_i, y_i)$, with $\ell_{CE}(\hat{y}_i, y_i)$ the pairwise cross-entropy loss between $\hat{y}_i$ and $y_i$. We define the machine translation loss (or seq2seq loss) from $x$ to $y$, $\mathcal{L}_{MT}$ as the sum of the negative log-likelihood of each token $y_i$, given $x$ and previous tokens $y_0 \ldots y_{i-1}$ (note that $x$ and $y$ can have different lengths) :

$$\mathcal{L}_{MT}(x, y) = -\sum_i \log\left(P(y_i|x, y_1 \ldots y_{i-1})\right)$$

### 3.1 COMMON OBJECTIVE FUNCTIONS

TransCoder (Roziere et al., 2020) learns to translate between programming languages by leveraging three unsupervised objectives developed for natural language (Lample et al., 2018b):

**Masked Language Modeling (MLM)** trains an encoder to predict randomly masked inputs. It is commonly used to pre-train embeddings for natural (Devlin et al., 2018; Liu et al., 2019) and programming languages (Kanade et al., 2020; Feng et al., 2020). MLM allows the model to learn the syntax and semantics of programs. Alternative objectives, have been proposed for programming languages (Guo et al., 2020; Lachaux et al., 2021; Ahmad et al., 2021; Wang et al., 2021). We do not use them here, as MLM remains effective and easy to use on a wide range of programming languages.

Denoting $mask(x)$ the masked version of the code sentence $x$, and $enc(t)$ the encoder output, MLM uses the following loss:

$$\mathcal{L}_{MLM} = \mathcal{L}_{CE}\left(enc(mask(x)), x\right). \tag{1}$$

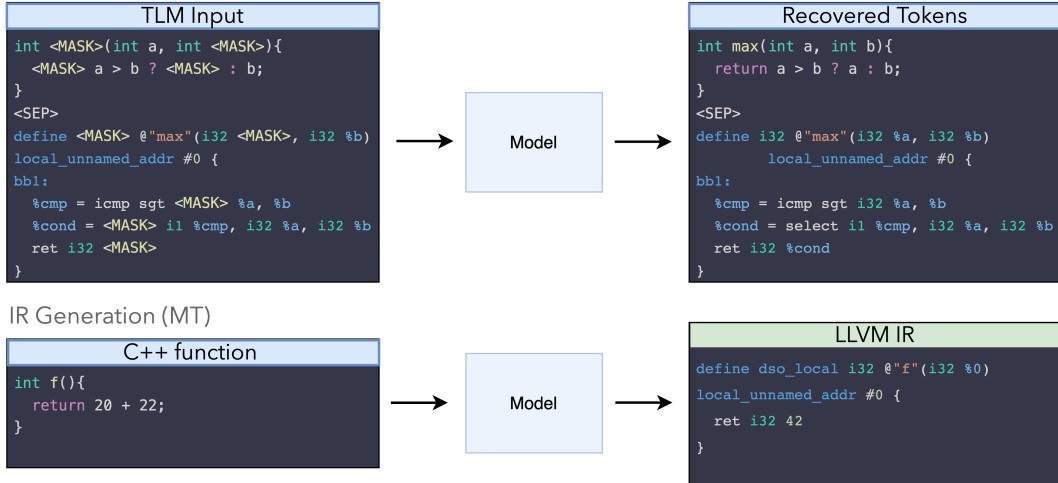

Figure 3: **IR for code representation objectives.** We show examples of masking (used in TLM and TAE) and IR generation used to improve code representations with IRs. The masking objective in TLM or TAE makes the model understand the relationship between code and IR. The IR generation objective helps the model to build semantic representations of the code. For instance, another C++ function computing `39 + 3` would result in the same IR. A Go function that returns 42 would also have a similar LLVM IR. Therefore, the IR Generation objective encourages the model to build similar representations for these three semantically equivalent functions.

**Denoising Auto Encoding (AE)** trains a sequence to sequence (seq2seq) model to retrieve an original sequence from a corrupted version. Corruption is done by masking spans of tokens randomly sampled from a Poisson distribution, as well as removing and shuffling tokens. It uses the following loss ($noise(x)$ denotes the corrupted version of $x$):

$$\mathcal{L}_{AE} = \mathcal{L}_{MT}\left(noise(x), x\right). \tag{2}$$

**Back-Translation (BT).** Back-Translation (Sennrich et al., 2015) uses the model to generate a noisy translation of the input sentence, and then trains the model to recover the original input from the translation. It is a simple yet powerful objective for unsupervised machine translation (Lample et al., 2018a; Artetxe et al., 2018). In practice, it is a required loss to get competitive performance, so it is a staple of all our experiments. Formally, we use the model to translate sequence $x$ into $\hat{y}$ and train the model to reverse the translation process, using the loss:

$$\mathcal{L}_{BT} = \mathcal{L}_{MT}\left(\hat{y}, x\right) \tag{3}$$

## 3.2 IR FOR CODE REPRESENTATIONS

Intermediate representations (IR) provide additional information about the code to be translated. We add them to the training dataset, as described in section 4.2, and leverage them by adding three new objective functions to those described in section 3.1.

**Translation Language Modeling (TLM),** first introduced in Lample and Conneau (2019), strives at generating common representations for parallel sentences in different languages. Like the masked language modeling (MLM) objective, it trains an encoder to predict random masked inputs. However, TLM is trained on pairs of parallel sentences, concatenated together and separated by a special token. Here, we concatenate functions in their source language and their corresponding IR, using the source code and IR language embeddings, and train the encoder to predict randomly masked tokens. This allows the model to learn correspondences between the source and the IR. The corresponding loss is ($\oplus$ denotes concatenation):

$$\mathcal{L}_{TLM} = \mathcal{L}_{CE}\left(mask(x \oplus z^{(x)}), x \oplus z^{(x)}\right) \tag{4}$$

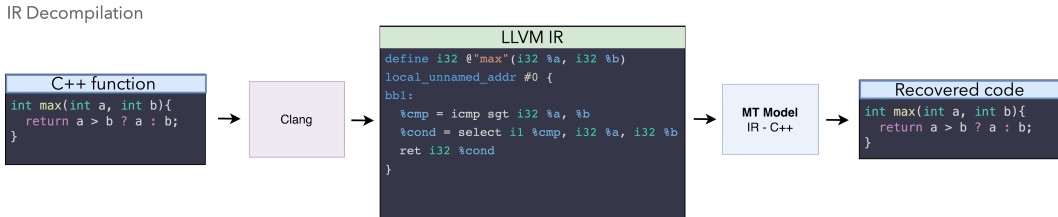

Figure 4: **IR Decompilation objective.** Here, we generate the IR corresponding to each function and train a model to decompile it. The IR pivot model uses this objective, as well as back-translation objectives, allowing it generalize to IRs generated from any language.

**Translation Auto-Encoding (TAE)**   amounts to transposing the TLM objective into a denoising auto-encoder. The source code and corresponding IR are corrupted and masked, and then concatenated into one sequence (using the language embeddings for code and IR, as previously). TAE is then tasked to recover the original, using the following loss:

$$\mathcal{L}_{TAE} = \mathcal{L}_{MT}\left(noise(x) \oplus noise(z^{(x)}), x \oplus z^{(x)}\right) \tag{5}$$

**IR Generation (MT)**   trains the model to translate the source code into the corresponding IR. This allows the encoder to learn source code representations from the semantics of the IR. The loss is:

$$\mathcal{L}_{IRGen} = \mathcal{L}_{MT}\left(x, z^{(x)}\right) \tag{6}$$

These three objectives need both the source code and the corresponding IR. However, only a fraction of the functions and files in our dataset could be compiled. To mitigate this, we also train the models on the full monolingual data using the MLM and AE objectives described above. In this setup, the back-translation (BT) objective is the same as in Roziere et al. (2020), and allows our model to translate directly from source code only at inference time.

### 3.3   ADDITIONAL LOSSES: IR DECOMPILATION AND PIVOT

We study two alternative uses of intermediary representations: IR decompilation, and IR pivot translation. IR decompilation consists of recovering source code corresponding to a given IR. In practice, it reverses the computations performed by the compiler. IR Pivot is a translation method built upon IR decompilation. Since LLVM can compile many languages (C++, Java, Rust, Go) into the same IR, an obvious approach to code translation consists of decompiling the IR generated from the source language into code in the target language. We call this method "IR pivot". Note that, whereas the IR for code representation techniques only used IR during training, both the decompilation and pivot method also need the IR for inference.

**Decompilation.**   In this supervised task, we use LLVM to generate IR from source code, and train a language model to reverse the process, i.e. learn to predict the source code from the IR. Models are pre-trained using the MLM and AE objectives, and decompilation is learned using the machine translation loss:

$$\mathcal{L}_{Decomp} = \mathcal{L}_{MT}\left(z^{(x)}, x\right) \tag{7}$$

**IR Pivot.**   This task leverages the IR as a pivot for code translation. For instance, to translate from Rust to C++, we first use LLVM to compile a Rust program into IR and then decompile the IR to C++ using a neural decompiler. In practice, slight variations exists between the IR generated for different languages: the Rust-IR and C++-IR behave like dialects of the LLVM-IR. This often leads to poor performance of the IR Pivot method. We mitigate these issues using a variety of techniques, which we describe in section C of the appendix.

## 4 DATA

### 4.1 TRAINING DATA

Our training data was extracted with Google BigQuery, which indexes over 2.8 million open source repositories from GitHub[2]. We selected projects whose license explicitly permits re-distribution of parts, and extracted all individual C++, Java, Rust and Go functions. To learn to decompile IRs, we also used the CodeNet dataset (Puri et al., 2021), a repository of 14 million competitive programming solutions in 55 languages. Our models work at function level: this reduces compilation failures over missing dependencies, while keeping sequence lengths short.

Table 1: **Dataset coverage across languages, in number of standalone functions.** More details can be found in Table 7 in the appendix.

|                          | C++      | Go       | Java    | Rust     |
|--------------------------|----------|----------|---------|----------|
| Monolingual data         | 6.6 M    | 9.4 M    | 7.8 M   | 576.3 K  |
| Code / IR Parallel Data  | 344.4 K  | 384.4 K  | 2.2 M   | 19.2 K   |
| Successful IR Compilation | 5.2%    | 4.1%     | 28.2%   | 3.3%     |

### 4.2 GENERATING INTERMEDIATE REPRESENTATIONS

While the LLVM ecosystem is large, not every language has an LLVM front-end, and not every front-end can produce LLVM IR out-of-the-box. We use `clang++`[3] Lattner and Adve (2004) from the established LLVM C++ compilation toolchain, JLang[4] for Java, Gollvm[5] for Go and `rustc` Matsakis and Klock II (2014) for Rust. For the same program, written in different languages, different front-ends may produce different IR. To minimize these variations, we process the source code as follows. First, we generate the most size-optimized IR (*-Oz* flag), which makes the IR more uniform across languages. Second, we strip all unnecessary information (e.g. header and footer with attributes, debug information, comments). Finally, block names are canonicalized and symbol names demangled to facilitate their recovery. The functions that fail to compile at this point (e.g. because of missing dependencies) are not included in the parallel dataset, as seen in the last row of Table 1.

### 4.3 EVALUATION

Traditional NMT evaluation relies on metrics such as BLEU, that are based on n-gram overlaps. However, when dealing with programming languages, syntax and in particular compilation and computation outputs can differ widely despite minor changes in the code. Conversely, semantically equivalent code, that differ only in variable names or order of operations can have a low BLEU score. To take this into account, we use and enhance the computational accuracy test suite from Roziere et al. (2020), that contains 852 parallel competitive programming solutions in C++, Java and Python. Using C2Rust, CxGo and some manual code cleaning, we translated 280 functions and test suites in Rust and 343 in Go to measure the performance of our models in these languages. We measure our performance using the computational accuracy (CA@1) metric (Kulal et al., 2019; Roziere et al., 2020), which considers that a translation is correct if it passes a series of unit tests.

## 5 RESULTS

### 5.1 EXPERIMENTAL DETAILS

For TransCoder, we consider a sequence-to-sequence (seq2seq) transformer model (Vaswani et al., 2017) with attention (Bahdanau et al., 2015; Sutskever et al., 2014) and the same architecture as Roziere et al. (2020). Our model has 12 layers (6 in the encoder and 6 in the decoder), 8 attention heads, and a dimension of 1024. For the objectives that add noise and masks to the input sentence, such as MLM, TLM, AE, and TAE, we choose the masked tokens and noise randomly on the fly at each epoch. We mask 15% of the tokens in MLM and TLM. In AE and TAE, we mask 20% of the tokens. MLM is trained on streams of data, while the other objectives are trained at function level. We

---

[2]`https://console.cloud.google.com/marketplace/details/github/github-repos`
[3]`https://clang.llvm.org/`
[4]`https://polyglot-compiler.github.io/JLang/`
[5]`https://go.googlesource.com/gollvm/`

Table 2: **Translation performance (CA@1), for greedy decoding and beam size 5.** "To X": average performance when translating to language X. "From X": average performance when translating from language X. See Table 3 in the appendix for more detailed results. All these methods except for the IR pivot also use the three objectives defined in TransCoder: MLM, DAE and Back-Translation (BT). All combinations of the TLM, MT and TAE objectives improve the performance compared to TransCoder. The best results are obtained when all three are used at the same time. Beam search, using beam size 5 and returning only the top element from the beam results in improved performance. The IR Pivot method generates a translation in the target language from an IR generated from the source, and performs poorly in our setting.

| | from C++ | to C++ | from Go | to Go | from Java | to Java | from Rust | to Rust | AVG |
|---|---|---|---|---|---|---|---|---|---|
| **Greedy decoding** | | | | | | | | | |
| IR Pivot | 17.4 | 24.0 | 19.9 | 11.5 | 11.9 | 22.2 | 16.3 | 7.8 | 16.4 |
| TransCoder (baseline) | 46.4 | 52.1 | 42.1 | 45.6 | 41.2 | 44.5 | 29.6 | 17.0 | 39.8 |
| TLM | 47.5 | 54.8 | 45.4 | 41.2 | 39.8 | 52.1 | 31.1 | 15.7 | 40.9 |
| MLM + TAE | 47.3 | 53.3 | **47.2** | 44.8 | 41.8 | 45.9 | 25.1 | 17.4 | 40.4 |
| TLM + TAE | 46.9 | **55.9** | 45.0 | 37.9 | 38.5 | **54.5** | 34.9 | 16.8 | 41.3 |
| MLM + MT | 45.5 | 51.0 | 44.0 | 48.9 | 46.6 | 45.2 | 25.7 | 16.6 | 40.5 |
| TLM + MT | 45.6 | 51.5 | 45.1 | 47.1 | 46.9 | 45.5 | 24.4 | 17.9 | 40.5 |
| TAE + MT | 47.8 | 54.3 | 43.8 | 43.9 | 39.1 | 49.2 | 33.4 | 16.7 | 41.0 |
| TLM + TAE + MT | **47.8** | 54.3 | 46.6 | **51.6** | **47.1** | 49.6 | **35.3** | **21.4** | **44.2** |
| **Beam size 5** | | | | | | | | | |
| TransCoder (baseline) | **53.8** | 53.4 | 45.2 | 54.4 | 46.1 | 51.5 | 35.9 | 20.9 | 45.3 |
| TLM + TAE + MT | 52.9 | **53.5** | **48.8** | **57.1** | **51.5** | **53.4** | **37.9** | **27.1** | **47.8** |

use the Adam optimizer (Kingma and Ba, 2015) and an inverse squared-root learning rate scheduler, with an initial learning rate of $10^{-5}$ in most of our experiments. Our models are implemented in PyTorch using mixed-precision floats. The pre-trained models were trained until convergence. The translation models presented in Tables 2 and 3 were trained for a week on 32 NVIDIA V100 GPUs.

## 5.2 IR-AUGMENTED CODE REPRESENTATIONS FOR TRANSLATION

Models using combinations of the three objectives—TAE, TLM and MT—introduced to leverage IR, were trained to translate between pairs of four languages (C++, Java, Rust, Go). Their average performance when translating to and from every language are presented in table 2. Additional information, including a comparison to TransCoder-ST for C++ ↔ Java, can be found in Table 3) in the appendix. As a baseline, we use a TransCoder (Roziere et al., 2020) model, trained with MLM on the same dataset.

Using greedy decoding, the new TLM, TAE and MT objectives, which leverage the IR, improve performance for every language. The best average results are obtained when combining all of them. Compared to TransCoder, they improve performance by an average 4.4% point (11% relative). The largest impacts are observed in the low data regime: translations from and into Rust (a language less represented in our training set) are improved by 25.6% and 19.3% (relative). Beam search improves the results of both TransCoder and our models, using IR-augmented representation still results in better performance. Qualitatively, we observe that IRs help our model translate types when the source and target types are represented by different tokens. For instance, in the first example of Table 1, it translates the semantics of `int` correctly using `i32` instead of an unsigned integer type (`usize`). See Appendix H for more analysis on how our objectives improve word embeddings.

Compared to IR-augmented translation models, the "obvious" IR Pivot method proves disappointing, even though it achieves non-trivial performances. It is heavily dependent on the size of the training set: the IR pivot performs relatively well when translating from low-resource to high-resource languages (e.g. from Rust), and badly when translating to low-resource languages (e.g. to Rust).

## 5.3 DECOMPILATION RESULTS

To compute the IR pivot, we trained a neural decompiler to retrieve source code from IRs. We tried two separate configurations for decompilation: a shared decoder with 6 layers for all language / IR pairs, or four separate decoders of with two layers each (one per language). Using a shared decoder

improves the performance for all languages, and particularly when the data is scarce (e.g. Rust). See Table 5 in the appendix for more information.

We compare the performance of our model to RetDec (Křoustek et al., 2017), a rule-based decompiler. It obtains a computational accuracy of 68.75 on our C++ dataset and a BLEU score of 8.54. In comparison, our model obtains a computational accuracy of 77.9 and a BLEU score of 63.6 in the same setting. In particular, RetDec fails to decompile LLVM files generated from C++ code, especially snippets leveraging the standard library structures such as $unordered\_map$ or $std::allocator$. The limitations of RetDec, which was implemented by a team of 24 developers in 7 years [6], shows how difficult it is to build exhaustive rule-based decompilers, especially when the IR comes from different languages or tools.

## 6 DISCUSSION

**Different IR and interpreted languages**  The four languages considered in this work have front-ends that can output LLVM Intermediary Representation. LLVM presently covers more than 30 computer languages. Using IR as pivot requires that the source and destination language have front-ends that use the same IR. This rules out some widely-used languages (e.g. Python). Using the IR to improve embeddings is less restrictive: the source and destination language can be trained on different IR, and aligned with back-translation. In this paper, we focus on compiled languages, but it is important to note that Intermediary Representations are usually available for interpreted languages as well: modern interpreters translate the source code into byte-code, that can serve as an IR.

**Pivot vs Embedding**  TransCoder is an unsupervised model that learns to align code representations and translate code from one language to another. It is based solely on source code and does not use IRs. The pivot method uses automatically generated parallel sentences to learn to decompile IRs, and back-translation to adapt to different IR dialects. This method learns to translate using only IR-level similarities, and does not use the source code itself except to compute the IR. Although it underperforms other methods, it performs relatively well when little data is available for the source language, because the IR can be computed using a rule-based compiler. However, it requires to compute IRs at test time, which can be cumbersome. Instead, adding the TLM, TAE, and MT objectives to the objectives generally used for unsupervised code translation allows the model to get the best of both worlds. It can learn multilingual representations of source code from similarities in the IR and in the source code itself. As shown in Table 2, it outperforms both TransCoder and the pivot method. At the same time, this model does not require to compute IRs at test time, and is as easy to use as TransCoder.

**Using our model at inference time.**  Our self-supervised IR-augmented TLM, TAE and MT objectives are designed to improve the multilingual code representations used in translation models. However, the translation task does not require to compute these objectives. Therefore, they lead to models that are just as simple to use as TransCoder: computing the IR is not required at test time and the model generates the translation directly from the source function.

## 7 RELATED WORKS

**Source-to-Source Translation.** Many rule-based methods are available for transpilation, an inventory of which can be found online[1]. In particular, C2Rust[7] and CxGo[8], along with manual corrections, were central for us in translating evaluation tests to Go and Rust (See Section 4.3). Similarly, 2to3[9], a Python library porting Python 2 code to Python 3, was used in Aggarwal et al. (2015) to create a parallel dataset and train a machine learning model.

Neural Machine Translation for code is hampered by the lack of parallel data between programming languages. Indeed, apart from a few language pairs, such as Java-C# (Nguyen et al., 2013; Chen

---

[6]https://blog.fpmurphy.com/2017/12/avast-retargetable-decompiler-ida-plugin.html
[7]https://github.com/immunant/c2rust
[8]https://github.com/gotranspile/cxgo
[9]https://docs.python.org/2/library/2to3.html

et al., 2018), and specific domains (e.g. competitive programming code), it is difficult to collect large datasets of semantically equivalent code in different languages. TransCoder (Roziere et al., 2020) bridges this gap by introducing unsupervised machine translation to programming languages. They take advantage of large monolingual code bases to learn to translate between C++, Python and Java with high performance. Later, DOBF (Lachaux et al., 2021) improved the model pre-training method used in TransCoder, and Roziere et al. (2022) used automatically generated unit tests to improve translation performance between Java, C++ and Python. Recently, large language models trained on code, such as Codex (Chen et al., 2021) and PALM (Chowdhery et al., 2022), have been used for unsupervised code translation.

Using the Transcoder model, Weisz et al. (2021) and Weisz et al. (2022) survey the links between humans and NMT methods for code translation. They view neural translation methods as aids to programmers. In this context, they demonstrate that even imperfect models can improve the quality of an engineer's work for code translation, and plead for the improvement of human-machine interfaces.

**Decompilation.** Like transpilation, decompilation is usually performed using rule-based methods that rely on pattern matching to parse the control flow structure of the program. RetDec, an open source decompiler created by Avast (Křoustek et al., 2017), can decompile an executable to C and a Python-like language via LLVM IR. Other tools exist, such as the Hex-Rays Decompiler[10] and Brumley et al. (2013). A thorough review of rule-based methods can be found in papers such as Liang et al. (2021a) and Katz et al. (2019). With these methods, decompilation can fail if the code is too convoluted, or if it contains language features that were not explicitly translated. Most methods also produce unstructured programs, relying on a large number of goto statements to simulate the control flow of the lower level programming languages. This is semantically correct, but very rarely found in human-written code.

A few works have studied the use of sequence-to-sequence neural networks for neural decompilation. Katz et al. (2019) uses LSTM networks to decompile LLVM IRs and assembly code to C. Their approach generates code templates based on the IR, that determine the structure of the output. Then, they fill them with correct variable assignments and numerical values. In the same vein, Fu et al. (2019) tries to address limitations of neural decompilation with two sequential phases: code sketch generation and iterative error correction. Finally, Liang et al. (2021b) use a method close to ours, and train Transformer models to translate between binary code and C.

**Intermediate representations** are almost as old as compiler design. The first IR, UNCOL (Strong et al., 1958) was introduced in the mid-1950s, together with the idea of reusing the same compiler for several languages and machines. In 1960, NELIAC (a variant of ALGOL) (Huskey et al., 1960) was the first retargetable compiler, portable to different architectures. Feldman (1979) describes how a compiler for Fortran 77 can be added to the C compilers of Johnson (1979) and Ritchie (1979). GCC (Stallman, 2001) introduces Register Transfer Language (RTL) a low-level IR inspired by Davidson and Fraser (1980), and then GENERIC and GIMPLE (Merrill, 2003), precursors of the IR used in LLVM (Lattner and Adve, 2004).

## 8 CONCLUSION

In this paper, we leverage LLVM IRs to improve neural machine translation for source code. The IR provides a common semantically-rich language, into which C++, Go, Java and Rust code can all be compiled. We develop three objectives, designed to leverage IRs for better multilingual representations of source code, which lead to a 5.5% relative average improvement for code translation. We also show that sequence-to-sequence transformers perform well for neural decompilation, and use this for pivot translation.

We only worked with the LLVM IR, but our approach is broadly applicable to any pair of languages that share a common Intermediate Representation. More generally any IR can help improve the code representations by tying them to the semantics. Another limitation is the scale of our current source and target sequences. As future work, LLVM IRs could be generated at a larger scale by compiling entire projects, which would greatly improve the percentage of successful IR compilations in Table 1. More languages and IRs could be used, and those extensions could be powered by larger models.

---

[10]https://hex-rays.com/decompiler/

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

## A  Full Scores Table

Table 3: **Results on unsupervised code translation.** The metric shown is the computational accuracy for a single generation (CA@1), measuring the translation correctness using unit tests. It is the full version of Table 2. The models were all trained with the same budget. As in Table 2, all these methods except for the IR pivot also use the three objectives defined in TransCoder: MLM, DAE and Back-Translation (BT). Although it is not the case for every language pair, TransCoder-IR, which uses the TLM, TAE, and MT objectives outperforms other methods on average. TransCoder-ST (Roziere et al., 2022) uses a parallel dataset generated with automated unit tests and outperforms other methods for C++ ↔ Java. Their method is orthogonal to ours, and we could also improve our performance with similar methods.

| | C++ → Go | C++ → Java | C++ → Rust | Go → C++ | Go → Java | Go → Rust |
|---|---|---|---|---|---|---|
| Baseline TransCoder | 57.7 | **63.3** | 18.2 | 56.1 | 46.9 | 23.3 |
| TransCoder-ST | - | 68.0 | - | - | - | - |
| Pivot | 16.1 | 22.0 | 14.0 | 30.5 | 26.5 | 2.7 |
| TLM | **61.8** | 62.5 | 18.2 | 57.6 | 56.4 | 22.2 |
| TAE | 57.7 | 62.5 | 21.7 | **63.0** | 54.7 | **23.8** |
| TLM + TAE | 58.2 | **63.3** | 19.2 | 55.2 | **57.0** | 22.8 |
| MT | 56.8 | 60.6 | 19.2 | 60.3 | 53.8 | 18.0 |
| TLM + MT | 58.6 | 58.5 | 19.7 | 57.3 | 54.1 | **23.8** |
| TAE + MT | 61.4 | 60.2 | 21.7 | 55.5 | 53.8 | 22.2 |
| TLM + TAE + MT | 55.9 | 62.9 | **24.8** | 61.8 | 55.7 | 22.2 |

| | Java → C++ | Java → Go | Java → Rust | Rust → C++ | Rust → Go | Rust → Java |
|---|---|---|---|---|---|---|
| Baseline TransCoder | 77.9 | 35.9 | 9.6 | 22.4 | 43.2 | 23.4 |
| TransCoder-ST | 84.6 | - | - | - | - | - |
| Pivot | 19.5 | 9.4 | 6.7 | 22.0 | 8.9 | 18.1 |
| TLM | 80.9 | 31.8 | 6.6 | 25.9 | 30.0 | 37.5 |
| TAE | 80.3 | 38.6 | 6.6 | 16.6 | 38.1 | 20.6 |
| TLM + TAE | **82.2** | 24.6 | 8.6 | **30.4** | 31.0 | **43.3** |
| MT | 76.2 | 50.9 | 12.6 | 16.6 | 39.1 | 21.3 |
| TLM + MT | 77.9 | **52.7** | 10.1 | 19.2 | 30.0 | 24.1 |
| TAE + MT | 77.5 | 33.6 | 6.1 | 30.0 | 36.6 | 33.7 |
| TLM + TAE + MT | 74.5 | 49.6 | **17.2** | 26.5 | **49.2** | 30.2 |

## B  BEAM SIZE EVALUATION

Table 4: **Results on unsupervised code translation with different beam sizes.** The metric shown is still the computational accuracy for a single generation (CA@1). BS N refers to beam search decoding with beam size N, and returning only the top element of the beam. Using beam search improves the average performance of every model. BS N means that the model is evaluated with beam size N. When the beam size is not given, we use greedy decoding. Surprisingly, beam size 5 outperforms beam size 10. Our method using intermediate representations still outperforms the baseline with beam size 5 and 10 in average. With the baseline, we obtain average CA@1 scores of 45.3 with beam size 5 and 44.0 with beam size 10. Our method yields CA@1 scores of 47.8 with beam size 5 and 46.8 with beam size 10.

| | C++ → Go | C++ → Java | C++ → Rust | Go → C++ | Go → Java | Go → Rust |
|---|---|---|---|---|---|---|
| Baseline TransCoder | 57.7 | 63.3 | 18.2 | 56.1 | 46.9 | 23.3 |
| Baseline TransCoder (BS 5) | **65.5** | 67.6 | 28.3 | 52.1 | **59.3** | 24.3 |
| Baseline TransCoder (BS 10) | 65.0 | **68.7** | 27.8 | 52.1 | 57.0 | 23.8 |
| TLM + TAE + MT | 55.9 | 62.9 | 24.8 | **61.8** | 55.7 | 22.2 |
| TLM + TAE + MT (BS 5) | 61.4 | 66.6 | **30.8** | 57.3 | 59.0 | **30.2** |
| TLM + TAE + MT (BS 10) | 61.4 | 67.4 | 29.3 | 56.4 | 59.0 | 29.1 |

| | Java → C++ | Java → Go | Java → Rust | Rust → C++ | Rust → Go | Rust → Java |
|---|---|---|---|---|---|---|
| Baseline TransCoder | 77.9 | 35.9 | 9.6 | 22.4 | 43.2 | 23.4 |
| Baseline TransCoder (BS 5) | **82.9** | 45.5 | 10.1 | 25.2 | **54.8** | 27.5 |
| Baseline TransCoder (BS 10) | 80.9 | 46.4 | 7.6 | 23.6 | 51.8 | 23.0 |
| TLM + TAE + MT | 74.5 | 49.6 | 17.2 | 26.5 | 49.2 | 30.2 |
| TLM + TAE + MT (BS 5) | 76.4 | **57.7** | **20.2** | **26.8** | 52.3 | **34.7** |
| TLM + TAE + MT (BS 10) | 77.7 | 57.3 | 18.2 | 26.2 | 51.8 | 28.2 |

## C  PIVOT METHOD DETAILS

As mentioned in Section 3.3, IR generated from different languages contain slight variations and can be seen as dialects of the same language. In practice, these variations prevent us from simply using our best decompilation model to generate source code in another language than the one used to generate the IR. Although we prompt the model to generate code in the target language with language embeddings, it learns to focus on the particularities of each dialect and ignores the language embeddings. Therefore, it generates code in the source language, which results in a computational accuracy score of 0 for translation.

One way to solve this issue is to use one decoder per target language. Then, the model is able to generate code in the target language. However, this method still performs poorly due to the small differences between the IR dialects. The method we tested that performed the best, and which is reported in Table 2, uses back-translation to make the model to translate from any IR dialect to any language. This model is also grounded by supervised translation steps making it generate IR from code and code from IR. In practice, we create new language embeddings for every IR dialect (i.e. IR-C++, IR-Go, IR-Java, IR-Rust) for depending on the source language. At training time, we make the model generate noisy translations in the IR-Go, IR-Java and IR-Rust "languages" for every C++ sequence, and train it to re-generate the C++ sequence from the noisy translation. To allow the model to generate good training data for IR-X → C++, we also generate noisy translations in Go, Java, and Rust for every IR generated from C++ in our dataset and train the model to retrieve the IR. Using our parallel code//IR dataset, we also train the model to translate between C++ and IR-C++ sequences. We do the same for every language and alternate between them.

# D IR DECOMPILATION

Table 5: **Performance of LLVM IRs Decompilation.** This table shows the computational accuracy (CA@1) of our neural decompiler and the RetDec C++ rule-based decompiler. Our neural decompiler outperforms RedDec on C++ and is more broadly applicable.

|  | C++ | Go | Java | Rust |
|---|---|---|---|---|
| Baseline - RetDec | 68.8 | — | — | — |
| Separate Decoders | 52.7 | 42.2 | 60.1 | 19.5 |
| Shared Decoder | **77.9** | **70.1** | **82.2** | **61.0** |

# E TRANSLATION EXAMPLES

| Input function (C++) | LLVM IR | Decompilation code (C++) |
|---|---|---|
| ```int max(int a, int b){``` 
 ```  for (int i = 0; i < 10; ++i)``` 
 ```    a = 2 * a - a;``` 
 ```  c = b``` 
 ```  return a - b + c > b ? a : b;``` 
 ```}``` | ```define i32 @"max"(i32 %a, i32 %b)``` 
 ```local_unnamed_addr #0 {``` 
 ```bb1:``` 
 ```  %cmp = icmp sgt i32 %a, %b``` 
 ```  %cond = select i1 %cmp, i32 %a, i32 %b``` 
 ```  ret i32 %cond``` 
 ```}``` | ```int max(int a, int b){``` 
 ```  return a > b ? a : b;``` 
 ```}``` |
| ```bool is_even(int n){``` 
 ```  return n & 1 == 0;``` 
 ```}``` | ```define dso_local zeroext``` 
 ```i1 @"is_even"(i32 %0)``` 
 ```local_unnamed_addr #0 {``` 
 ```  ret i1 false``` 
 ```}``` | ```bool is_even(int n){``` 
 ```  return false``` 
 ```}``` |
| ```long multiplyBySeven(long n){``` 
 ```  return ((n << 3) - n);``` 
 ```}``` | ```define dso_local``` 
 ```i64 @"multiplyBySeven"(i64 %0)``` 
 ```local_unnamed_addr #0 {``` 
 ```  %2 = mul i64 %0, 7``` 
 ```  ret i64 %2``` 
 ```}``` | ```long multiplyBySeven(long n){``` 
 ```  return n * 7;``` 
 ```}``` |

Figure 5: **Code simplification examples with Decompilation / Pivot.** Since the LLVM IR is optimized, functions that are semantically equivalent after optimization map to the same IR. In the first example, it allows to remove useless code by decompiling the generated LLVM IR. In the second example, the simplification allows to find a bug: the `&` operator has precedence over `==` in C++, causing this function to always evaluate to `false`. It is not obvious when looking at the input code, but becomes clear with the IR and simplified C++ code. In the third example, it replaces a bitwise operation by a more straightforward multiplication. In all examples, we can run the compiler again to check that the IR of the decompiled code is exactly the same as that of the input. It guarantees that the input and simplified code have the same semantics.

| Input function | TransCoder | TransCoder-IR |
|---|---|---|

```java
// Java
static int addOne ( int x ) {
  return ( - ( ~ x ) ) ;
}
```

```rust
// Rust
fn add_one (x : int ) -> int {
    (- ( ~ x) as int)
}
```

```rust
// Rust
fn add_one (x : i32 ) -> i32 {
    (- ( ! x))
}
```

```java
// Java
static boolean isEven (int n){
  return ( n % 2 == 0 );
}
```

```rust
// Rust
fn is_even ( n : u32 ) -> bool {
    ( n % 2 == 0 ) as bool
}
```

```rust
// Rust
fn is_even ( n : i32 ) -> bool {
    ( n % 2 == 0 )
}
```

Figure 6: **Java to Rust translation examples.** In the first example, the IR allows the model to understand that the Java bitwise complement operator ~ should be replaced by ! in Rust. Also, it allows the model to translate the type correctly in both examples and avoids unnecessary casts. The IR allows the model to generate the right types (e.g. i32 instead of u32 when translating int) and operator (e.g. ! instead of ~ in Rust).

| Input function | TransCoder-IR |
|---|---|

```rust
// Rust
pub fn binary_search(nums: Vec<i32>,
                     target: i32,
) -> i32 {
    if nums.is_empty() {
        return 0
    }
    if target < nums[0] {
        return 0
    }
    let mut left = 0;
    let mut right = nums.len() - 1;
    while left <= right {
        let mid = left + (right - left) / 2;
        if nums[mid] == target {
            return mid as i32;
        } else if nums[mid] < target {
            left = mid + 1;
        } else {
            right = mid - 1;
        }
    }
    return left as i32;
}
```

```go
// Go
func binarySearch(nums[] int, target int) int {
    if len(nums) == 0 {
        return 0
    }
    if target < nums[0] {
        return 0
    }
    left: = 0
    right: = len(nums) - 1
    for left <= right {
        mid: = left + (right - left) / 2
        if nums[mid] == target {
            return mid
        } else if nums[mid] < target {
            left = mid + 1
        } else {
            right = mid - 1
        }
    }
    return left
}
```

Figure 7: **Rust to Go translation example.** This function performs binary search to find the insertion index for an element in an ordered vector. The model translates types, function definitions, variable definitions, and while loops correctly.

| Input function | TransCoder-IR |
|---|---|

```go
// Go
func pairsInSortedRotated(arr[] int,
                          n int,
                          x int)
int {
  var i int
  for i = 0; i < n - 1; i++{
    if arr[i] > arr[i + 1] {
      break
    }
  }
  var l int = (i + 1) % n
  var r int = i
  var cnt int = 0
  for l != r {
    if arr[l] + arr[r] == x {
      cnt++
      if l == (r - 1 + n) % n {
        return cnt
      }
      l = (l + 1) % n
      r = (r - 1 + n) % n
    } else if arr[l] + arr[r] < x {
      l = (l + 1) % n
    } else {
      r = (n + r - 1) % n
    }
  }
  return cnt
}
```

```cpp
// C++
int pairsInSortedRotated(int arr[],
                         int n,
                         int x)
{
  int i;
  for (i = 0; i < n - 1; i ++) {
    if (arr [i] > arr [i + 1]) break;
  }
  int l = (i + 1) % n;
  int r = i;
  int cnt = 0;
  while (l != r) {
    if (arr [l] + arr [r] == x) {
      cnt ++;
      if (l == (r - 1 + n) % n) return cnt;
      l = (l + 1) % n;
      r = (r - 1 + n) % n;
    }
    else if (arr[l] + arr[r] < x) l = (l + 1) % n;
    else r = (n + r - 1) % n;
  }
  return cnt;
}
```

Figure 8: **Go to C++ translation example.** This function computes the number of pairs of elements that sum to a given target in a sorted and rotated array. Our TransCoder-IR model translates it correctly.

| Input function | TransCoder-IR |
|---|---|

```rust
// Rust
pub fn get_max_repetitions(s1: String,
                           n1: i32,
                           s2: String,
                           n2: i32,
) -> i32 {
    if n1 == 0 {
        return 0;
    };
    let mut indices = vec![0;n1 as usize + 1];
    let mut counts = vec![0;n1 as usize + 1];
    let mut index = 0;
    let mut count = 0;
    for i in 1.. = n1 as usize {
        for j in 0..s1.len() {
            if (
                s1.as_bytes()[j]
                == s2.as_bytes()[index]
            ) {
                index += 1;
            }
            if index == s2.len() {
                index = 0;
                count += 1;
            }
        }
        counts[i] = count;
        indices[i] = index;
        for k in 0..i {
            if indices[k] == index {
                let pre_count = counts[k];
                let pattern_count = (n1 - k as i32)
                    / (i - k) as i32
                    * (counts[i] - pre_count);
                let remain_count = counts
                    [
                        k
                        + (n1 as usize - k)
                        % (i - k)
                    ] - pre_count;
                return (
                    pre_count
                    + pattern_count
                    + remain_count
                ) / n2;
            }
        }
    }
    counts[n1 as usize] / n2
}
```

```go
// Go
func getMaxRepetitions(s1 string,
                       n1 int,
                       s2 string,
                       n2 int,
) int {
    if n1 == 0 {
        return 0
    };
    indices := make([] int, n1 + 1)
    counts := make([] int, n1 + 1)
    index := 0
    count := 0
    for i := 1;
    i <= n1;
    i++{
        for j: = 0;
        j < len(s1);
        j++{
            if s1[j] == s2[index] {
                index++
            }
            if index == len(s2) {
                index = 0
                count++
            }
        }
        counts[i] = count
        indices[i] = index
        for k := 0;
        k < i;
        k++{
            if indices[k] == index {
                preCount := counts[k]
                patternCount := int(n1 - k)
                    / (i - k)
                    * (counts[i] - preCount)
                remainCount := counts[
                    k + (n1 - k) % (i - k)
                ] - preCount
                return (
                    preCount
                    + patternCount
                    + remainCount
                ) / n2
            }
        }
    }
    return counts[n1] / n2
}
```

Figure 9: **Rust to Go translation example.** We call S1 the string n1 repeated s1 times and S2 the string n2 repeated s2 times. This function finds the largest number of repetitions of S2 appearing in any subset of S1. The model translates the types correctly, understands that casting vector indices to unsigned int (i.e. with as usize) is not required in Go, and correctly translates other Rust constructs to Go.

# F  DATASET SIZE DETAILS

Table 6:  **Dataset details: number of tokens in our function-level dataset. This dataset contains only functions defined outside of classes and static functions.**

|  | Number of tokens | Number of sentences |
|---|---|---|
| **Monolingual data** | | |
| C++ | 2.33B | 6.6M |
| Go | 1.9B | 9.4M |
| Java | 1.5B | 7.8M |
| Rust | 130.0M | 576.3K |
| **Code / IR Parallel Data** | | |
| C++-IR | 946.7M | 343.9K |
| Go-IR | 971.8M | 384.4K |
| Java-IR | 1.7B | 2.2M |
| Rust-IR | 77.7M | 19.4K |

# G  ADDITIONAL ABLATIONS

**Training on IRs with different objectives.**   We perform some additional ablations to determine whether our performance improvements come from training on IRs or from our TLM, TAE and MT objectives. When training a model with the three objectives of TransCoder (i.e. MLM, DAE and BT) and considering the IR as an extra language, we obtain an average computational accuracy of $37.4$, which is lower than that of our baseline TransCoder. As the structure of the IR is not similar to that of any of our source languages, there is not much to gain from adding the IR as an extra language. Moreover, the model is wasting some time to compute the AE and BT objectives for the IR which can be better spent on the source languages. It confirms that our objectives are required to map IRs and their corresponding source code to similar representations in embedding space.

**Language ablation: no Java.**   As Rust and Go are more similar to Java than to C++, we also train a baseline model on C++, Go and Rust only to evaluate whether including Java hurts the translation performance. We observed similar performance for C++ $\leftrightarrow$ Go. However, we also observe a clear decrease in performance in the very low data regime (i.e. when translating to or from Rust). The computational accuracy for Rust $\rightarrow$ C++ goes down from $22.4\%$ to $20.1\%$ and it goes down from $43.15\%$ to $32.5\%$ for Rust $\rightarrow$ Go.

# H  WORD EMBEDDINGS

We notice that our method generates improved word embeddings. It is visible when looking at the cosine similarity for embeddings of rust types and their equivalents in C++. For instance, Figure 10 shows that the embedding of `u32` from our model leveraging LLVM IRs is most similar to `uint32` (with a cosine similarity of 0.4869). `uint`, which is also a correct translation, comes in $11^{th}$ position with a cosine similarity (0.3716). In contrast, `u32` has a similarity of only 0.2828 with `int`. This token, which would be an incorrect translation, comes only in $29^{th}$ position.

The baseline model, which does not use the IR, learns similar representations for rust types since they appear in similar contexts. Hence, its embedding of `u32` is most similar to other rust types tokens such as `u64`, `i32` or `u16`. `uint32` comes only in fourth position with a cosine similarity of 0.4218. Moreover, `uint` and `int` have almost the same cosine similarities with `u32` with the baseline model. It causes the model to often confuse unsigned and signed integer types, and to incorrectly translate `u32` into `int` instead of `uint`.

```
Baseline TransCoder:              Our model with TLM + TAE + MT:
1 - 0.4789 - u64                  1 - 0.4869 - uint32
2 - 0.4478 - i32                  2 - 0.4779 - u64
3 - 0.4223 - u16                  3 - 0.4681 - u16
4 - 0.4218 - uint32               4 - 0.4302 - uint32_t
5 - 0.4119 - usize                5 - 0.4270 - i32
...                               ...
14 - 0.3169 - uint                11 - 0.3716 - uint
...                               ...
16 - 0.3108 - int                 29 - 0.2828 - int
```

Figure 10: **Token similarities.** Rank and token similarity with `u32` for our model (right) and the baseline model (left). Our model generates embeddings that better capture token semantics.

## I  ANALYSIS OF ERROR TYPES

Table 7: **Rust error types.** To validate our intuition on the usefulness of the IR representations to decrease the number of type-related errors (see Fig.1 or Fig.6), we perform an in-depth analysis of the types of errors encountered for the Java → Rust direction. Here we count the total number of errors (there can be several for a single translation). We notice that the number of type-related errors (excluding E0433, E0425 and Others) decreases by 24% (609 vs. 463) and the number of mismatched types decreases by 49%.

| Error Code | Error Description | Baseline (Transcoder) | TLM + TAE + MT |
|---|---|---|---|
| E0308 | Mismatched Type | 414 | 210 |
| E0412 | Type Does Not Exist | 15 | 3 |
| E0277 | Type has Missing Trait | 180 | 250 |
| E0425 | Undefined Variable | 18 | 27 |
| E0433 | Use of Undefined Crate, Module or Type | 15 | 32 |
| — | Others | 28 | 33 |
| **TOTAL** | — | 670 | 555 |

