# OpenReview forum: "Code Translation with Compiler Representations"
_ICLR.cc/2023/Conference — ICLR 2023 notable top 25%_

### Official Review · Reviewer_DPxt · 2022-10-19

**Confidence:** 5
**Correctness:** 4
**Technical Novelty And Significance:** 4
**Empirical Novelty And Significance:** 4
**Recommendation:** 10

**Clarity, Quality, Novelty And Reproducibility:**

The paper is clear and reproducible.

A few minor comments:

* Writing could be more straightforward - it took me 5 sections to understand the main approach. I would move the current Section 5 to appear right after the introduction. The Related Work section can be deferred to the end of the paper.
* Minor: I am guessing that it is technically difficult and requires non-trivial dependencies to compile the programs to LLVM. Please release your pretrained models and detailed instructions on how to re-train them.
* Figure 4 is small and difficult to read.



**Strength And Weaknesses:**

## Strengths
* The approach is simple and provides strong empirical results in unsupervised translation over TransCoder.
* The paper basically transforms the "unsupervised translation problem" into a "supervised translation into a proxy language": Instead of learning an unsupervised translation between `A<->B`, the authors transform the task into a **supervised** translation between each of `A` and `B` languages to `LLVM` (that is, `A<->LLVM` and `B<->LLVM`). That is, they transform the unsupervised translation problem into a supervised translation to a proxy language.
The LLVM language serves as a proxy that every language can be **supervisedly** translated to,
which is very clever and very novel in my opinion.
* The main Section 5 is very clear.
* At test time, the model does not require additional analysis / conversion / intermediate representation of the input program, which is a huge advantage compared to prior work that relies on compiler information. Much such prior work which leverages compiler/runtime information - requires that information at test time as well.
* The evaluation is convincing and thorough, evaluating 4 languages and many objective ablations.

## Weaknesses
* Nothing serious, see the next section

## Questions:
Would it be meaningful to run the following ablation: the standard objectives (as TransCoder), but also include the LLVM data of the source and target languages (train on the LLVM as additional, separate examples from their C++/Java/Go original programs)?

The reason I am asking this is because the proposed approach is different from TransCoder in two aspects:
1. Trains on LLVM as well
2. Slightly different objectives TLM/TAE/MT.

What other experiments could disentangle these two contributions?


**Summary Of The Paper:**

The paper addresses code translation. While learning to translate from language `A` to language `B` (i.e., `A<->B`), the paper proposes to also train on translating `A<->LLVM` and `B<->LLVM`.
This way, LLVM serves as a proxy language that both `A` and `B` translate to.
At test time, no intermediate language is required, and the model translates directly `A<->B`.
This way, the `A<->LLVM` and `B<->LLVM` training serves as an effective training time supervision.
The approach is demonstrated on the task of unsupervised translation, with 4 programming languages, and shows significant improvements over TransCoder.

**Summary Of The Review:**

The paper is very clear, easy to read, and shows strong results over TransCoder using a simple and elegant approach.

The higher-level idea of transforming the "unsupervised translation" problem into a "supervised translation to a proxy language" is very novel and clever.

The proposed approach is appealing since the LLVM information it requires can be generated from the existing training data.
Even if not all data can be compiled to LLVM, the authors show that a small amount of such supervised LLVM data is sufficient.
Further, at test time, the trained model can be evaluated without any additional information.

Overall this is an excellent paper and I would be happy to see it accepted.

---

> ### Author Response · Authors · 2022-11-12
> **Response to Reviewer DPxt**
>
> We thank you for your feedback and insight. Please find the answers to your questions below:
>
> > Would it be meaningful to run the following ablation: the standard objectives (as TransCoder), but also include the LLVM data of the source and target languages (train on the LLVM as additional, separate examples from their C++/Java/Go original programs)?
> > The reason I am asking this is because the proposed approach is different from TransCoder in two aspects:
> > 1. Trains on LLVM as well
> > 2. Slightly different objectives TLM/TAE/MT.
>
> We understand your concern. It is true that training on a new language (such as the IR) can also impact the performance for the other languages. For instance, one could imagine that adding a language with some constructs similar to those of Java and others similar to those of C++ would help the model to represent those constructs and translate.
>
> Here, the pre-trained MLM model we use was actually trained on the available data from all languages *and* all the available LLVM IRs. We also experimented with a MLM model trained only on source languages and did not observe any significant difference without using the IR for finetuning with the MT and TAE objectives. It only improved our performance when using the IRs for finetuning.
>
> We are also ran an experiment to evaluate if training on the IR at finetuning time improves our performance.
> We obtain an average computational accuracy of 37.4, which is lower than that of our baseline TransCoder. It shows that using parallel data with objectives such as ours is a crucial factor in our ability to improve model performance using LLVM IRs.
>
> > The paper is clear and reproducible.
> > A few minor comments:
> > * Writing could be more straightforward - it took me 5 sections to understand the main approach. I would move the current Section 5 to appear right after the introduction. The Related Work section can be deferred to the end of the paper.
>
> Thank you for your feedback. We agree with your suggestions. In the revised version, we moved Section 5, which now appears earlier in the paper. We moved the related work section to the end of the paper as you suggested. We kept a small section presenting intermediate representations in compilers between the introduction and the training objectives section (former Section 5), as we believe it is useful to give some context on IRs before presenting the objectives.
>
> > Minor: I am guessing that it is technically difficult and requires non-trivial dependencies to compile the programs to LLVM. Please release your pretrained models and detailed instructions on how to re-train them.
>
> We are planning to release our pre-trained models, test datasets, and our code after the review period. We will also add detailed instructions on how to re-create the training dataset and re-train our models.
>
> > Figure 4 is small and difficult to read.
>
> Thank you for your feedback. We made the text in Figure 4 larger in order to make it easier to read.

---

### Official Review · Reviewer_d6WQ · 2022-10-24

**Confidence:** 4
**Correctness:** 3
**Technical Novelty And Significance:** 3
**Empirical Novelty And Significance:** Not applicable
**Recommendation:** 6

**Clarity, Quality, Novelty And Reproducibility:**

The paper is well written and easy to follow. I note a few typos below. The investigation showing the deficiency of the IR pivot objective and the proposed remedies are sufficiently novel. As far as I could tell, the work does not offer to release its dataset, which (at least, the IR portion) is likely very challenging to collect, so reproducing this work would be very difficult.

Typos:
- P7: "consists in" -> "consists of"
- P8: "non trivial performances" -> "non-trivial performance"


**Strength And Weaknesses:**

The main strength of this work is the observation that IR can be made useful for unsupervised program translation by extending the conventional unsupervised translation objectives to include program IR, _rather than_ attempting to map programs into and from IR directly (what the paper calls "IR Pivot"), which tends to yield poor performance. As such, the conceptual contribution is somewhat straightforward (adding several loss terms based on concatenating IR and code), but nevertheless novel and significant, especially when considering the rather substantial amount of effort invested to collect hundreds of thousands of Code/IR pairs.

The main weakness, to me, lies in the evaluation. For one, the work is somewhat unclear in its presentation of the results. The introduction to Section 5 states that all six objectives are used during training, which presumably means both the three TransCoder and three new objectives. That would lead me to read Table 2, from the "TLM" row onwards, as including all three TransCoder objectives by default, but some rows also include "MLM". Does that mean that no row here represents the combination of all six components, and that the "TLM" row works rather well using just a single objective? Furthermore, given that TLM and TAE are very similar, I would like to see the results of "TAE + MT" given that "TLM + MT" is presented. There are a few other issues here:
- The "to Java" result has the wrong number bold-fonted (should be: "TLM + TAE: 54.5");
- I would expect Katz'19 to be included in the comparison since it is cited as a closely related, if older, technique;
- It is quite surprising that beam-size 5 yields _worse_ performance for "to C++" than using greedy decoding (may be worth double-checking this number);
- The "79%" improvement of Java to Rust mentioned in the abstract appears to be relative to an MLM baseline (Tab. 3, App. A), not TransCoder. Is this supposed to be the full TransCoder model? If not, it seems off to cite this as such a large improvement given that the baseline is realtively weak compared to the de facto standard model, even if TransCoder did not evaluate with this setting.

Please aim to improve the above results. Given that the score improvement over TransCoder is modest on average (and quite small for some languages), it is important for judging the significance of this work that the results given are reliable.

**Summary Of The Paper:**

This work proposes to improve automated program translation from unsupervised data by leveraging compiler intermediate representation (IR), a machine-code like format that is unified across many programming languages. Rather than train a model to translate each language into and from this domain, the work argues for using intermediate representation to augment inputs at training time in order to learn better representations, and omit it during inference. The result is a tool that performs somewhat better than a baseline without IR (TransCoder) without requiring the use of a compiler for inference.

**Summary Of The Review:**

The proposed use of IR to improve unsupervised program translation is reasonably effective and goes beyond the naive (and ineffective) approach of using IR as a proxy for an intermediate domain for universal machine translation. The results are largely convincing, but suffer from a number of smaller issues that should be remedied to make the paper stronger.

---

> ### Author Response · Authors · 2022-11-12
> **Response to Reviewer d6WQ 1/2**
>
> We thank you for your feedback and insight. Please find the answers to your questions below:
>
> > The main weakness, to me, lies in the evaluation. For one, the work is somewhat unclear in its presentation of the results. The introduction to Section 5 states that all six objectives are used during training, which presumably means both the three TransCoder and three new objectives. That would lead me to read Table 2, from the "TLM" row onwards, as including all three TransCoder objectives by default, but some rows also include "MLM". Does that mean that no row here represents the combination of all six components, and that the "TLM" row works rather well using just a single objective?
>
> Thank you for your comment. We agree that the way we present the results in this table can be confusing. The rows with “TLM” actually use both MLM and TLM. We believe that keeping the MLM objective is mostly useful because it allows us to pre-train on the data for which we were not able to compute IRs automatically. It means that the last two rows correspond to using all six objectives at the same time. We clarify it by removing the reference to MLM in the leftmost column of Table 2 (and Table 3 in the appendix). We also add a sentence in the caption, reminding that all the methods except for the IR pivot also use the MLM, DAE and Back-Translation objectives. We also use similar notations to clarify Tables 4 and 5 in the Appendix.
>
> > Furthermore, given that TLM and TAE are very similar. I would like to see the results of "TAE + MT" given that "TLM + MT" is presented.
>
> We followed your suggestion and added the scores for TAE + MT in Table 2 and Table 3.
>
> > The "to Java" result has the wrong number bold-fonted (should be: "TLM + TAE: 54.5");
>
> You are absolutely right. Thank you very much for noticing our mistake! We corrected it in the revised version.
>
> > I would expect Katz'19 to be included in the comparison since it is cited as a closely related, if older, technique;
>
> Katz’19 is related to our work, as it studies the use of LSTM networks to decompile LLVM IRs. However, there are some important differences, that make it difficult to compare their results to ours:
> 1) They decompile only LLVM IRs generated from C code while we study C++, Java, Rust and Go.
> 2) They generate random programs from pre-defined grammars, while we use real open-source code. Hence the distribution of their train and test datasets differs significantly from ours. Moreover, our own experiments using code generators to generate large amounts of code/IR pairs did not lead to improvements for neural decompilation on real code or code translation.
> 3) Their work targets neural decompilation by itself, while we aim to improve neural networks for code translation.
>
> > It is quite surprising that beam-size 5 yields worse performance for "to C++" than using greedy decoding (may be worth double-checking this number);
>
> We checked our results and confirm that the performance for “to C++” is worse when using beam size 5 than when using greedy decoding. As shown on Table 5 in the Appendix, it is due to beam search being worse for Go -> C++ and only marginally better for Java -> C++ and Rust -> C++. Here, we consider only the top element of the beam. Despite returning sequences with higher log-likelihood (according to the model), beam search does not always lead to improved generations.
>
> > The "79%" improvement of Java to Rust mentioned in the abstract appears to be relative to an MLM baseline (Tab. 3, App. A), not TransCoder. Is this supposed to be the full TransCoder model? If not, it seems off to cite this as such a large improvement given that the baseline is realtively weak compared to the de facto standard model, even if TransCoder did not evaluate with this setting.
>
> We believe that our notations in this table were confusing, and we clarify them. Thank you for bringing this to our attention.
> “MLM baseline” meant using MLM for pre-training a model trained with denoising auto-encoding (DAE) and Back-Translation (BT). It has exactly the same objectives as in TransCoder and we just retrained the model on different languages.
>
> > The paper is well written and easy to follow. I note a few typos below. The investigation showing the deficiency of the IR pivot objective and the proposed remedies are sufficiently novel.
>
> Thank you for noticing these typos. We fixed them in the revised version.

---

> > ### Author Response · Authors · 2022-11-12
> > **Response to Reviewer d6WQ 2/2**
> >
> > > As far as I could tell, the work does not offer to release its dataset, which (at least, the IR portion) is likely very challenging to collect, so reproducing this work would be very difficult.
> >
> > We plan to release our test dataset. Our train dataset was created from raw data downloaded from Google BigQuery. We plan to release the sql command we used to download it, as well as the scripts we used and clear instructions on how to generate the IRs.
> >
> > > The results are largely convincing, but suffer from a number of smaller issues that should be remedied to make the paper stronger.
> >
> > Thank you for your feedback. We hope we fixed the smaller issues you mentioned. Please don’t hesitate to tell us if you have more concerns or questions.

---

> > > ### Comment · Reviewer_d6WQ · 2022-11-16
> > > **Re: Response to Reviewer d6WQ**
> > >
> > > I thank the authors for their clarifications and corresponding edits; the paper is improved because of them. I continue supporting its acceptance. Regarding the beam search results, it might be good to emphasize that the "beam size 5" setting refers to using only the top candidate following beam search, as this is often used synonymously with top-5 accuracy (or at least, I associated the two when reading the results). It makes sense that it doesn't always improve results in the top beam configuration.

---

> > > > ### Author Response · Authors · 2022-11-18
> > > > **Second response to reviewer d6WQ**
> > > >
> > > > We thank you for reviewing our paper, and making suggestions that allowed us to improve it! In the new revision of the paper, we also emphasize that beam size 5 refers to using only the top element from the beam in order to avoid any confusion.

---

### Official Review · Reviewer_ikso · 2022-10-24

**Confidence:** 5
**Correctness:** 3
**Technical Novelty And Significance:** 2
**Empirical Novelty And Significance:** 3
**Recommendation:** 5

**Clarity, Quality, Novelty And Reproducibility:**

I think the empirical part of the evaluation is novel. However, it requires some in-depth study (please see the weakness part).
Methodology wise, the paper is rather weak.

**Strength And Weaknesses:**

Strength:

+ The paper is generally well-written and tries to address an interesting problem. The augmentation of IR during translation is a direct way to incorporate structures and some semantics into the code representation, and IR is generated by the compiler. This significantly cuts down the search space and improves the outcome.

+ The paper evaluates low-resourced languages like Go and Rust.

+ IR is also not needed for inference time.

+ They propose alternative design and decompilation as another application.

Weakness:

- The paper is not super novel. It essentially shows augmenting data from a different information source helps to improve NMT. However, I also believe compilation-ware IR augmentation is a good direction toward programming language translation.

- The evaluation is not in-depth. For example, in Tables 2 and 3, different settings (TLM, TAE, and MT) proved to be effective for different settings. There are no explanations for such behavior. Why certain objective functions are more effective in one language translation than another?

- Low-resourced languages like Go and Rust are more similar to C/C++ than Java. I am wondering whether including Java can hurt their translations, especially for translating C++ to Rust and C++ to Go and vice versa. It would be good to understand the impact of different language designs on translation.






**Summary Of The Paper:**

For programming language translation tasks (e.g., C# to Rust), the paper augments the source language with intermediate language representation (IR) generated by LLVM. They show that such IR-augmented translation can significantly improve the SOTA.
They also showed two alternate design choices where IR is used for neural decompilation and as a pivot task.

**Summary Of The Review:**

The paper shows Intermediate Representation (IR) augmentation to source code can improve NMT. While this is a good direction to improve programming language translation, novelty-wise, it is rather thin. The empirical evaluation also requires some in-depth study.

---

> ### Author Response · Authors · 2022-11-12
> **Response to Reviewer ikso**
>
> We thank you for your feedback and insight. Please find the answers to your questions below:
>
> >The paper is not super novel. It essentially shows augmenting data from a different information source helps to improve NMT. However, I also believe compilation-ware IR augmentation is a good direction toward programming language translation.
>
> We understand your concern. We believe that the novelty of our paper lies in:
> 1) Using compiler representation (here LLVM IRs) to improve code representations
> 2) Using IRs for code translation
> 3) Evaluating three novel objectives, using the IR at train time while not requiring to compute any IRs at test time.
>
> We also believe that compilation-aware IR augmentation could be used for other programming tasks.
>
> > The evaluation is not in-depth. For example, in Tables 2 and 3, different settings (TLM, TAE, and MT) proved to be effective for different settings. There are no explanations for such behavior. Why certain objective functions are more effective in one language translation than another?
>
> We have some qualitative analyses of the improvements provided by our system in Figure 1 and in Appendix E. We also discuss our results in Section 6.2. We believe that our objectives act similarly. TLM and TAE are very similar objectives. The main difference is that TLM is used to pre-train models while TAE is used afterwards. The IR generation (MT) objective forces the model to build semantic representation of source code, allowing it to generate the IR. We have seen these objectives work especially well for language pairs with frequently used semantics being represented by different tokens (e.g. `i32` in rust vs `int` in C++ and java).
>
> We also add some analysis of the embeddings obtained with our method and the baseline at the end of the appendix.
>
> > Low-resourced languages like Go and Rust are more similar to C/C++ than Java. I am wondering whether including Java can hurt their translations, especially for translating C++ to Rust and C++ to Go and vice versa. It would be good to understand the impact of different language designs on translation.
>
> Our model uses trainable language embeddings, which help it to adapt its representations to the source or target language. We believe that adding Java does not hurt the performance for other languages.
>
> We are running this experiment. We will add the results in the appendix of the paper.
>
> Edit: we added it in the appendix.
> We train a baseline model on C++, Go and Rust only to evaluate whether including Java hurts the translation performance. We observed similar performance for C++ ↔ Go. However, we also observe a clear decrease in performance in the very low data regime (i.e. when translating to or from Rust). For instance, the computational accuracy for Rust → C++ goes down from 22.4% to 20.1% and it goes down from 43.15% to 32.5% for Rust → Go.

---

### Official Review · Reviewer_2BBL · 2022-10-26

**Confidence:** 4
**Correctness:** 4
**Technical Novelty And Significance:** 3
**Empirical Novelty And Significance:** 3
**Recommendation:** 5

**Clarity, Quality, Novelty And Reproducibility:**

__Clarity and Reproducibility__
- More details should be provided regarding the pretraining setup. Just mentioning models were trained for a week on 32 NVIDIA V100 GPUs is not enough. For reproducibility, it is important to mention the amount of data used (number of tokens, GB of data, number of training steps, etc.).
- The paper mentions, "unsupervised machine translation consists of two tasks: training language embeddings (one embedding per language) and aligning them" - what does training language embeddings mean? Is it truly an indispensable part of unsupervised MT?
- I am unsure if equations 3, 4, and 5 are needed. They are too simple to be required to emphasize.

__Quality and Novelty__
- While the use of IR for PL translation seems new, the scientific contribution of this work is thin. The paper reads more as describing a system rather than a research paper.
- I felt the paper could be compressed and presented as a short paper. It seems the use of unnecessary illustrations, simple math equations, and details of well-known methods are added to span 9 pages.


**Strength And Weaknesses:**

__Strengths__
- The main proposal of the work is well-motivated. The related work discussions are very comprehensive.
- The use of IR is interesting and seems new. While the idea seems straightforward, the underlying effort to build such a system proposed in this work is significant.
- The paper extended an existing translation dataset to new languages with test cases.

__Weaknesses__
- The writing of the method section could be improved. It became confusing with the mentions of the TransCoder pretraining objectives and the methods used in this work (sections 5.1 and 5.2).
- Table 1 shows the percentage of successful IR compilation is small. Is it worth pursuing such an approach to building PL translation systems?
- In Table 2, rows 2 - 7 in the greedy decoding block (starting from TransCoder) show the performance difference across models is very small, but row 8 shows TLM + TAE + MT performs quite well. It is not clear why?
- While the empirical performances show improvements, there is not enough analysis to emphasize what makes the proposed approach truly effective.
- In the introduction, the paper mentions, "Augmenting training data with the corresponding IR can benefit a Neural Transpiler in two ways: it helps align embeddings for different languages and improves the semantic understanding of the code." I do not see the authors validating this statement.
- TransCoder has subsequent improvements in the form of DOBF and TransCoder-ST models. Why they were not mentioned in the empirical results?



**Summary Of The Paper:**

The paper takes an interesting approach to training neural models for programming language (PL) translation - use of Intermediary Representations (IR): language-agnostic pseudocode that describes the semantics of the program. The paper proposes to augment code translation with IRs, specifically LLVM IR, to facilitate translations between C++, Java, Rust, and Go languages. The main idea of training neural sequence-to-sequence models relies on pretraining via translation language modeling (TLM), translation auto-encoding (TAE), and IR generation via machine translation (MT). The paper extends a prior dataset of parallel functions to Go and Rust languages. The experiment results show improvements in PL translation over the prior approaches.

**Summary Of The Review:**

I would not prefer to see this paper accepted at the ICLR conference. The paper reads more like a technical report that describes a system using IR, which is interesting. The paper could go into a system conference or on a different track other than the main research track in an ML conference like ICLR. I am unsure if this paper passes the bar of a research paper of ICLR. However, I am not absolutely certain about my judgment, therefore, I am ready to be convinced otherwise.

---

> ### Author Response · Authors · 2022-11-12
> **Response to reviewer 2BBL 1/2**
>
> We thank you for your feedback and insight. Please find the answers to your questions below:
>
> > The writing of the method section could be improved. It became confusing with the mentions of the TransCoder pretraining objectives and the methods used in this work (sections 5.1 and 5.2).
>
> We clarify this by stating clearly that all the objectives used in TransCoder are also used here. We now mention this clearly in the captions of the result tables and clarifying that MLM is used in all of them.
>
> > Table 1 shows the percentage of successful IR compilation is small. Is it worth pursuing such an approach to building PL translation systems?
>
> We believe that the low percentage of successful IR compilation shows that there is still room for improvement. The size of the dataset could be increased substantially with more engineering effort to generate LLVM IRs for more files and projects.
>
> Our approach does not require to generate IRs at test time and brings positive results despite the small number of files we compiled. It shows that this approach is worth pursuing.
>
> > In Table 2, rows 2 - 7 in the greedy decoding block (starting from TransCoder) show the performance difference across models is very small, but row 8 shows TLM + TAE + MT performs quite well. It is not clear why?
>
> All our objectives seem to bring some gains compared to the baseline and using all three provides larger gains in our experiments. We also added a line for TAE + MT (pre-trained with MLM instead of TLM), which also performs well.
>
> > While the empirical performances show improvements, there is not enough analysis to emphasize what makes the proposed approach truly effective.
>
> We have several qualitative analyses of the improvements provided by our system in Figure 1 and Appendix E. They provide some insight into why our approach is effective. For instance, it shows that our model is much better at translating types correctly when they use different tokens in different languages. It is particularly impactful in Rust, where `int` types are often incorrectly translated to `usize` (unsigned integer) instead of `i32`. It is also better when translating tokens that are close or the same but have different meanings. For instance, it does not confuse `>` with `>>` in Figure 1 and translates the Java token `~` into the rust token `!` correctly in Figure 6.
>
> As you suggested below, we also added more analyses on the embeddings we obtain with our method and the vanilla TransCoder objectives.
>
> > In the introduction, the paper mentions, "Augmenting training data with the corresponding IR can benefit a Neural Transpiler in two ways: it helps align embeddings for different languages and improves the semantic understanding of the code." I do not see the authors validating this statement.
>
> Our objectives are designed to better capture token semantics. Moreover, they contribute only by improving token embeddings, as the objectives which allow to generate code in the target language (i.e. denoising auto-encoding and back-translation) remain unchanged.
> We agree that the improvements to the embeddings could be shown more concretely in the paper. Therefore, we added some data and analysis at the end of the appendix (Appendix G) showing how our objectives lead to improved embeddings compared to using only the original TransCoder objectives.
>
> > TransCoder has subsequent improvements in the form of DOBF and TransCoder-ST models. Why they were not mentioned in the empirical results?
>
> DOBF is not shown in the empirical results because it was trained only on Python and Java, while we train on C++, Java, Rust and Go in this work.
>
> TransCoder-ST was included in Table 3 in the Appendix for C++ -> Java and Java -> C++, as it was not trained on Rust and Go.
>
> The improvements in DOBF and TransCoder-ST are orthogonal to those we present here. The model we train here could also be used to generate parallel datasets with automatically generated unit tests like in TransCoder-ST.
>
> > More details should be provided regarding the pretraining setup. Just mentioning models were trained for a week on 32 NVIDIA V100 GPUs is not enough. For reproducibility, it is important to mention the amount of data used (number of tokens, GB of data, number of training steps, etc.).
>
> We added more information about the training dataset in Table 6 at the end of the appendix. In this work, back-translation steps are significantly slower than other types of updates, as they require to generate translations auto-regressively. Therefore, we believe that comparing models with a fixed time budget is preferable.

---

> > ### Author Response · Authors · 2022-11-12
> > **Response to reviewer 2BBL 2/2**
> >
> > > The paper mentions, "unsupervised machine translation consists of two tasks: training language embeddings (one embedding per language) and aligning them" - what does training language embeddings mean? Is it truly an indispensable part of unsupervised MT?
> >
> > Thank you for your feedback, we acknowledge that this sentence is unclear and modify it in the revised paper. It now writes "Unsupervised machine translation consists of learning multilingual sequence embeddings, and generating sequence in any output language from these embeddings".
> >
> > Unsupervised machine translation learns aligned embeddings, which are similar for code snippets with similar semantics in different languages. It is truly indispensable, as it is what allows the model to generate translations with the same semantics as the input. The denoising auto-encoding and back-translation are enough to learn such embeddings, but pre-training the model using the MLM objective leads to substantial performance gains as shown in the DOBF paper. For instance, they obtain a CA@1 of 44.8 for Python -> Java with MLM pre-training while it is only 24.0 without any pre-training.
> >
> > > I am unsure if equations 3, 4, and 5 are needed. They are too simple to be required to emphasize.
> > > I felt the paper could be compressed and presented as a short paper. It seems the use of unnecessary illustrations, simple math equations, and details of well-known methods are added to span 9 pages.
> >
> > We understand that equations 3, 4 and 5 are probably unnecessary for some of our readers who are already familiar with similar objectives. However, we know from first-hand feedback that these equations significantly facilitated the understanding of our objectives for some of our readers and also received some positive feedback about it. We prefer to keep these equations to make our paper more accessible.
> >
> > While we agree that our paper could be compressed, we believe that compressing it into a much shorter version would make it much more difficult to understand (at least for some readers). The appendix would also become disproportionately large compared to the main paper.
> >
> > > While the use of IR for PL translation seems new, the scientific contribution of this work is thin. The paper reads more as describing a system rather than a research paper.
> >
> > Our paper is the first exploring the use of compiler representations to improve machine learning models for tasks on programming languages. We believe that the system we describe answers this research question and shows some potential for further exploitation of compiler representations for code translation or other tasks.

---

> > ### Comment · Reviewer_2BBL · 2022-11-16
> > **Reply to authors comments**
> >
> > > The improvements in DOBF and TransCoder-ST are orthogonal to those we present here. The model we train here could also be used to generate parallel datasets with automatically generated unit tests like in TransCoder-ST.
> >
> > If we have methods like TransCoder-ST, why do we need yet another technique/paper? Performance wise TransCoder-ST is better than the proposed method of this work. TransCoder-ST could be adopted for Rust and Go languages too. I assume authors wanted to say, technique wise TransCoder-ST and IR-based methods are orthogonal but they are techniques for the same downstream task. So, we need to compare them and bring it in the main body of the paper. But the authors perhaps included TransCoder-ST in the Appendix, so that we do not complain about inferior performance of the proposed IR-based approach.
> >
> > > We have several qualitative analyses of the improvements provided by our system in Figure 1 and Appendix E. They provide some insight into why our approach is effective. For instance, it shows that our model is much better at translating types correctly when they use different tokens in different languages. It is particularly impactful in Rust, where int types are often incorrectly translated to usize (unsigned integer) instead of i32. It is also better when translating tokens that are close or the same but have different meanings. For instance, it does not confuse > with >> in Figure 1 and translates the Java token ~ into the rust token ! correctly in Figure 6.
> >
> > Qualitative examples are good but they are not enough to characterize overall improvements. How difficult it is to quantify that the proposed method reduces type-related errors in translation? For example, we could do error analysis as in [1] (see table 7 in the Appendix). Currently, all analysis in the paper are qualitative, a few anecdotal examples cannot justify the improvements achieved by the proposed approach.
> >
> > [1] Summarize and Generate to Back-translate: Unsupervised Translation of Programming Languages

---

> > > ### Author Response · Authors · 2022-11-18
> > > **Response to further comments**
> > >
> > > > If we have methods like TransCoder-ST, why do we need yet another technique/paper? Performance wise TransCoder-ST is better than the proposed method of this work. TransCoder-ST could be adopted for Rust and Go languages too. I assume authors wanted to say, technique wise TransCoder-ST and IR-based methods are orthogonal but they are techniques for the same downstream task. So, we need to compare them and bring it in the main body of the paper. But the authors perhaps included TransCoder-ST in the Appendix, so that we do not complain about inferior performance of the proposed IR-based approach.
> > >
> > > TransCoder-ST was only used on high-resource programming languages (i.e. Java, C++ and Python) and relies on being able to train a performant unsupervised translation model to start generating verified translations. The method we present here performs well for training such a model, especially when the types are difficult to translate (e.g. when translating to Rust). With extra work to generate multilingual unit tests in Rust and Go on top of Java and C++, we could use our models to kickstart a TransCoder-ST for Rust and Go and generate larger parallel datasets.
> > >
> > > We believe that the comparison of our model with TransCoder is fair because the starting conditions for ST and IR are orthogonal and thus those approaches are not comparable although they are complementary. We included TransCoder-ST in Table 3 in Appendix A and linked to it in the main paper in order to help reviewers and readers compare their results to ours easily. We did not include it in Table 2 because metrics for TransCoder-ST are only available for C++ <-> Java, while Table 2 presents average results to and from each language. However, we understand your point and agree that this comparison could be missed. We add a sentence in the main paper stating that a comparison to TransCoder-ST can be found in Table 3.
> > >
> > > > Qualitative examples are good but they are not enough to characterize overall improvements. How difficult it is to quantify that the proposed method reduces type-related errors in translation? For example, we could do error analysis as in [1] (see table 7 in the Appendix). Currently, all analysis in the paper are qualitative, a few anecdotal examples cannot justify the improvements achieved by the proposed approach.
> > > > [1] Summarize and Generate to Back-translate: Unsupervised Translation of Programming Languages
> > >
> > > We thank you for your feedback and agree that a quantitative analysis is also valuable in this case. We add one in the appendix of the paper. Among other things, it shows a 49% reduction in the “mismatched type” type of error when translating from Java to Rust. It confirms that our objectives help the model translate types to Rust correctly and our qualitative analysis on embeddings shows how it happens.

---

### Author Response · Authors · 2022-11-12
**Rebuttal Highlights**

We thank all the reviewers for their feedback and suggestions. We added some data to the paper and modified it. We believe that the paper is now clearer and easier to read thanks to the feedback we received. Thank you! Most of the modifications we made are highlighted in blue in the revision.
* We clarified the objectives used in Tables 2, 3 and 4. We write that every objective except for the pivot uses MLM, DAE and BT and remove the “MLM” mention in some of the model names (as this objective is used in every model)
* We added an appendix (G) with some analysis on how our novel objectives improve code representations
* We added the scores for the TAE + MT objective to Tables 2 and 3. We are also running other experiments requested by reviewers ikso and DPxt. We will add them in the appendix or in the main paper
* We plan to release our pre-trained models, extended test dataset, and our code. We will also add clear instructions on how to re-create our train dataset, re-train a model and reproduce our results
* We corrected some typos noticed by the reviewers. We also reorganized the sections following the advice of reviewer DPxt: the training objectives section (previous Section 5) now appears much earlier as Section 3 and the related work section was moved to the end of the paper

---

### Decision · Program_Chairs · 2023-01-20

**Decision:**

Accept: notable-top-25%

**Justification For Why Not Higher Score:**

It is of interest that adding this auxiliary information improves the code translation problem, and of NMT in general is of broad interest to the ICLR community.  The level of importance that might suggest oral presentation is reduced by the fact that the availability of such auxiliary information is limited to the more narrowly scoped case of program translation.

**Justification For Why Not Lower Score:**

Good novelty, significant improvement over other methods.

**Metareview: Summary, Strengths And Weaknesses:**


The paper addresses the automated translation of programs from one source language to another (e.g. C++ <-> Java <-> go <-> Rust).
It improves on the state of the art on this problem by augmenting training by incorporating LLVM IR, which is easily obtained.  IR is not required at test time.

All reviewers consider the approach novel and effective, and that the experiments convincingly demonstrate the improvements obtained.

Some reviewers consider the paper too verbose, some find it harder to understand - the authors have clearly given thought to their exposition, and taken some of the reviewers' comments into account.  In my view, the paper is well pitched.

There is some question as to how to fairly compare to Transcoder-ST, for the high-resource Java-C++ case.   The comparison in Table 3 is important to include (and is now called out in the revision).    The Transcoder-ST results are not currently presented in bold -- that would be important to fix, or perhaps to use a different color.

The authors suggest they will release the dataset as a bigtable query.  If this is in order to avoid redistribution issues, I would consider it nevertheless very important to release enough additional information that practitioners can verify they have the same data, e.g. a checksum for each file/fragment in the results.

**Note From Pc:**

if the above contains the word "oral" or "spotlight" please see: "oral" presentation means -> notable-top-5% and "spotlight" means -> notable-top-25%. As stated in our emails, we are disassociating presentation type from AC recommendations